# Quantifying Anonymity in Score-Based Generators with Adversarial Fingerprinting

## Abstract

Recent advances in score-based generative models have led to a huge spike in the development of downstream applications using generative models ranging from data augmentation over image and video generation to anomaly detection. Despite publicly available trained models, their potential to be used for privacy-preserving data sharing has not been fully explored yet. Training diffusion models on private data and disseminating the models and weights rather than the raw dataset paves the way for innovative large-scale data-sharing strategies, particularly in healthcare, where safeguarding patients' personal health information is paramount. However, publishing such models without individual consent of, *e.g.*, the patients from whom the data was acquired, necessitates guarantees that identifiable training samples will never be reproduced, thus protecting personal health data and satisfying the requirements of policymakers and regulatory bodies. This paper introduces a method for estimating the upper bound of the probability of reproducing identifiable training images during the sampling process. This is achieved by designing an adversarial approach that searches for anatomic fingerprints, such as medical devices or dermal art, which could potentially be used to uniquely re-identify training images. Our method harnesses the learned score-based model to estimate the probability of the entire subspace of the score function that may be utilized for one-to-one reproduction of training samples. To validate our estimates, we generate images containing a fingerprint and investigate whether generated samples from trained generative models can be uniquely mapped to the original training samples. Overall our results show that privacy-breaching images can be reproduced at sampling time if the models were trained without care.

## 1 Introduction

Maintaining privacy and anonymity is of utmost importance when working with personal identifiable information, especially if data sharing has not been individually consented and thus cannot be shared with other institutions (Jin et al., 2019). The potential of incorporating privacy preserving methods into the training to allow sharing of mathematically equal synthetic datasets derived from private datasets would be significant and could potentially solve many problems, including racial bias (Larrazabal et al., 2020) and the difficulty of applying techniques such as robust domain adaptation (Wang et al., 2022). Recent advances in generative modeling, *e.g.*, effective diffusion models (Song et al., 2020a; Dhariwal & Nichol, 2021; Rombach et al., 2022; Ruiz et al., 2022), enabled the possibility of model sharing as shown by Pinaya et al. (2022). However, it remains unclear to what extent a shared model reproduces training samples and whether or not this raises privacy concerns.

In a general, our focus centers on the notion of utilizing a dataset labeled as $D$ of samples from the image distribution $p_{data}(\mathbf{x})$. Then the goal is to train a generative model $s$, which learns only on private data. Direct privacy breaches would occur if the generative model exhibits a non-zero probability for memorizing and reproducing samples from the training set.

Guarantees that such privacy breaches will not occur would ultimately allow to train models based on proprietary data and share the models instead of the underlying data sets. Healthcare providers would be able to share complex patient information like medical images on a population basis instead of needing to obtain individual consent from patients, which is often infeasible, especially retrospectively. Guarantees that no personal identifiable information is shared would furthermore

pave the way to population studies on a significantly larger scale than currently possible and allow to investigate bias and fairness of downstream applications on anonymous distribution models of sub-populations.

However, currently trained and published models can be prompted to reproduce training data at sampling time. Somepalli et al. (2023) have observed that diffusion models are able to reproduce training samples and Carlini et al. (2023) have even shown how to retrieve faces of humans from training data, which raises serious privacy concerns. Other generative models are directly trained for memorization of training samples (Cong et al., 2020). In a medical setting, it remains uncertain whether merging segments from various images truly poses any privacy risks. Therefore, we propose a scenario with an adversarial that has some prior information about a training sample and would consequently be able to filter out the image based on this information. In medical imaging this could be any medical device, a skin tattoo, an implant, or heart monitor; any detectable image with visual features that are previously known. Then an attacker could generate enough samples and filter images until one of the generated samples contains this feature. If the learned marginal distribution of the generative model that contains this feature is slim, then all images generated with it will raise privacy concerns. We will refer to such identifiable features as *fingerprints*. To estimate the probability of reproducing fingerprints, we propose to use synthetic anatomical fingerprints (SAF), which can be controlled directly through synthetic manipulations of the training dataset and reliably detected in the sampling dataset. In summary our main contributions are:

- We formulate a realistic scenario in which unconditional generative models exhibit privacy problems due to the potential of training samples being reproduced.
- We introduce a mathematical approach to determine the maximum probability of producing sensitive data, from which we derive a readily calculable indicator metric.
- We evaluate this indicator by computing it for different datasets and show evidence for its effectiveness.
- We show that anomalies in the training set of diffusion models are either memorized or forgotten but never augmented
- We investigate three real world examples in which this indicator could be successfully used to evaluate diffusion models.

## 2 BACKGROUND

Consider $D$ containing samples from the real image distribution $p_{data}(\mathbf{x})$. In general, highly effective generative methods like diffusion models (Rombach et al., 2022) work by modeling different levels of perturbation $p_\sigma(\tilde{\mathbf{x}}) := \int p_{data}(\mathbf{x})p_\sigma(\tilde{\mathbf{x}} \mid \mathbf{x})\mathrm{d}\mathbf{x}$ of the real data distribution using a noising function defined by $p_\sigma(\tilde{\mathbf{x}} \mid \mathbf{x}) := \mathcal{N}(\tilde{\mathbf{x}}; \mathbf{x}, \sigma^2\mathbf{I})$. In this case $\sigma$ defines the strength of the perturbation, split into N steps $\sigma_1, \dots, \sigma_N$. The assumption is that $p_{\sigma_1}(\tilde{\mathbf{x}} \mid \mathbf{x}) \sim p_{data}(\mathbf{x})$ and $p_{\sigma_N}(\tilde{\mathbf{x}} \mid \mathbf{x}) \sim \mathcal{N}(\mathbf{x}; \mathbf{0}, \sigma_N^2\mathbf{I})$

Then we can define the optimization as a score matching objective by training a model $\mathbf{s}_{\boldsymbol{\theta}}(\mathbf{x}, \sigma)$ to predict the score function $\nabla_{\mathbf{x}} \log p_\sigma(\mathbf{x})$ of the noise level $\sigma \in \{\sigma_i\}_{i=1}^N$.

$$\boldsymbol{\theta}^* = \arg\min_{\boldsymbol{\theta}} \sum_{i=1}^N \sigma_i^2 \mathbb{E}_{p_{\mathrm{data}}(\mathbf{x})}\mathbb{E}_{p_{\sigma_i}(\tilde{\mathbf{x}}|\mathbf{x})}\big[\, \|\mathbf{s}_{\boldsymbol{\theta}}(\tilde{\mathbf{x}}, \sigma_i) - \nabla_{\tilde{\mathbf{x}}}\log p_{\sigma_i}(\tilde{\mathbf{x}} \mid \mathbf{x})\|_2^2 \,\big]. \tag{1}$$

For sampling, this process can be reversed, for example, using Markov chain Monte Carlo estimation methods following Song & Ermon (2019). Song et al. (2020b) extended this approach to a continuous formulation by redefining the diffusion process as a process given by a stochastic differential equation (SDE):

$$\mathrm{d}\mathbf{x} = \mathbf{f}(\mathbf{x}, t)\mathrm{d}t + g(t)\mathrm{d}\mathbf{w}, \tag{2}$$

and training a dense model on predicting the score function for different time steps t, where $\mathbf{w}$ models the standard Wiener process, $\mathbf{f}$ the drift function of $\mathbf{x}(t)$, that models the data distribution,

$g(t)$ is the diffusion coefficient, and $\mathbf{x}(t)$ the drift coefficient. Therefore, the continuous formulation of the noising process, denoted by $p_t(\mathbf{x})$ and $p_{st}(\mathbf{x}(t) \mid \mathbf{x}(s))$, is used to characterize the transition kernel from $\mathbf{x}(s)$ to $\mathbf{x}(t)$, where $0 \leq s < t \leq T$. Anderson (1982) showed that the reverse of this diffusion process is also a diffusion process. The backward formulation is

$$\mathrm{d}\mathbf{x} = [\mathbf{f}(\mathbf{x}, t) - g(t)^2 \nabla_{\mathbf{x}} \log p_t(\mathbf{x})]\mathrm{d}t + g(t)\mathrm{d}\bar{\mathbf{w}}, \tag{3}$$

Finally, Song et al. (2020b) show that the reverse diffusion process of the SDE can be modeled as a deterministic process as the marginal probabilities can be modeled deterministically in terms of the score function. As a result, the problem simplifies to an ordinary differential equation:

$$\mathrm{d}\mathbf{x} = \left[\mathbf{f}(\mathbf{x}, t) - \frac{1}{2}g(t)^2 \nabla_{\mathbf{x}} \log p_t(\mathbf{x})\right]\mathrm{d}t, \tag{4}$$

and can therefore be solved using any black box numerical solver such as the explicit Runge-Kutta method. This means that we can perform exact likelihood computation, which is typically done in literature (Song et al., 2020b), to estimate how likely the generation of test sample, *e.g.* images, is. This means that low negative log-likelihood (NLL) is desirable. In our case, we want to estimate the likelihood of reproducing training samples at test time. Ideally, this probability would be zero or very close to zero.

## 3 METHOD

Typically, NLL measures how likely generating test samples at training time is. To use it to evaluate the memorization of training data, it is possible to compute the NLL of the training dataset. A limitation of using NLL is that it only computes the likelihood of the exact sample to be reproduced at sampling time and therefore is insufficient for giving estimates of the likelihood of generating samples that raise privacy concerns. We can compute the likelihood of the exact sample but this does not mean that the images in the immediate neighborhood are not leading to privacy issues. To resolve this issue, we propose to estimate the upper bound of the likelihood of reproducing samples from the entire subspace that belongs to the class of private samples. First, we define the sample $\mathbf{x}_p$ that we consider to be a potential privacy breach and augment this sample by adding a synthetic anatomic fingerprint (SAF) to it. This SAF is used to identify the sample, which raises privacy concerns. Then we repeatedly apply the diffusion and reverse diffusion process for different noise levels and check when the predicted sample starts to diverge to a different image.

### 3.1 ESTIMATION METHOD

Let $p_s(\mathbf{x}_p)$ define the likelihood of the model $\mathbf{s}$ to reproduce the private sample $\mathbf{x}_p$ at test time. Following Eq. (4), we can compute the likelihood of this exact sample. However, this does not account for the fact that images in the immediate neighborhood, like slightly noisy versions of $\mathbf{x}_p$, are not anonymous. Consequently, we are interested in computing $q(p)$, which is defined as the likelihood of reproducing any sample that is similar enough to the target image that it raises privacy concerns:

$$q(p) = \int_{\Omega_p} p_s(\mathbf{x})\mathrm{d}\mathbf{x}, \tag{5}$$

where $\Omega_p$ is defined as the region of the image $\mathbf{x}_p$ that is private. We determine this region by training a classifier tasked with detecting whether the image belongs to the image class, as explained in Sec. 3.4. To search through the image manifold, we make use of the reverse diffusion process centered around the SAF image $\mathbf{x}_p$ defined as $p_{t,b} := p(\mathbf{x}_t \mid \mathbf{x}_p) = \mathcal{N}(\tilde{\mathbf{x}}; \mathbf{x}_p, \sigma_t^2 \mathbf{I})$ for $\mathbf{x}(s)$ to $\mathbf{x}(t)$, where $0 \leq t \leq T$. We can employ the diffusion process centered around this image to sample from the neighborhood and then use the learned reverse diffusion process to generate noisy samples $\mathbf{x}_{t,p}$. Then we can use this as starting image for the reverse diffusion process to sample $\mathbf{x}'_{t,p}$:

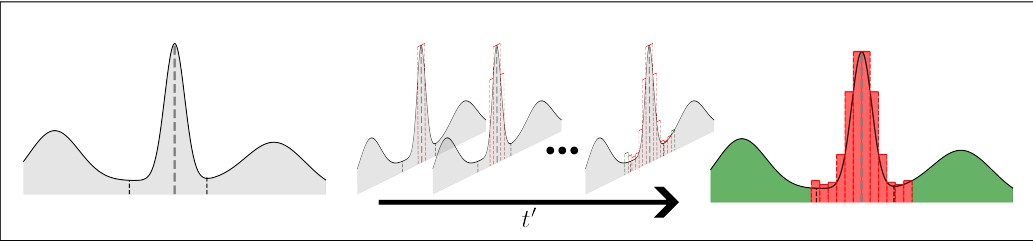

Figure 1: Illustration of our estimation method in 1D. The grey line denotes the query image $\mathbf{x}_p$. The estimation method iteratively increases the search space in the latent space of the generative model. The green area corresponds to image regions resulting in non-privacy concerning generated samples, while the red area is considered critical.

$$q(p) = \int_{\Omega_p} p_s(\mathbf{x})\mathrm{d}\mathbf{x} \approx \int_0^{t'} p_s(\mathbf{x}_{t,p})\mathrm{d}\mathbf{t} = \int_0^{t'} \mathbb{E}_{p(\mathbf{x}_{t,p})}\big[p(\mathbf{x}'_{t,p})\big]\mathrm{d}\mathbf{t}. \tag{6}$$

Technically, we could employ exact likelihood computation to estimate $q(p)$ but this would require integrating over the continuous image-conditioned diffusion process, which would be intractable in practice. Therefore, we propose to approach and estimate this integral by computing the Riemann sum of this integral and give an upper bound estimate for it using the upper Darboux sum:

$$\int_0^{t'} \mathbb{E}_{p(\mathbf{x}_{t,p})}\big[p(\mathbf{x}'_{t,p})\big]\mathrm{d}\mathbf{t} = \sum_t (\sigma_t - \sigma_{t-1})\mathbb{E}_{p(\mathbf{x}_{t,p})}\big[p(\mathbf{x}'_{t,p})\big] \leq \tag{7}$$

$$\sum_{i=0}^{t'} \sup_{t \in [t_i, t_{i+1}]} (\sigma_{t_{i+1}} - \sigma_{t_i})\mathbb{E}_{p(\mathbf{x}_{t,p})}\big[p(\mathbf{x}'_{t,p})\big], \tag{8}$$

which approaches the real value for steps that are small enough. We can compute this value by using $\mathbf{x}_p$ as a query image and estimating the expectation by performing Monte-Carlo sampling but this would be computationally infeasible due to the complexity of exact likelihood estimation.

### 3.2 METHOD INTUITION

Intuitively, we model the image space using the learned distribution of the score function $\nabla_{\tilde{\mathbf{x}}} \log p_{\sigma_i}(\tilde{\mathbf{x}} \mid \mathbf{x})$ by reversing the diffusion process and checking when the model starts to "break out" by generating images classified as different samples. For large $t$, the learned marginals $p(\mathbf{x}, t)$ span the entire image space. Importantly, by definition of the diffusion process, the distribution approaches the same distribution as the sampling distribution of the diffusion process if $\sigma_t$ gets large enough $p_{\sigma_N}(\tilde{\mathbf{x}} \mid \mathbf{x}_p) \sim \mathcal{N}(\mathbf{x}; \mathbf{0}, \sigma_N^2 \mathbf{I})$. However, for lower $t$ the model has learned that the distribution collapses towards a single training image $\mathbf{x}_p$. Essentially, it has modeled part of the subspace as a delta distribution around $\mathbf{x}_p$. We want to find out how far back in the diffusion process we have to go for the model to start to produce different images. The boundary $\Omega_p$ is then defined as all images that would collapse towards this training image and estimated using the classifiers. Fig. 1 illustrates this process in one dimension. Note that this is different from simply defining a variance that is large enough for the classifiers to fail, as $s_\theta(\mathbf{x}_p, \sigma_t)$ was trained to revert this noise.

### 3.3 SYNTHTETIC ANATOMIC FINGERPRINT

Let $\tilde{D}$ be our real dataset of size $N$ without any privacy concerns due to the lack of any identifiable information. Then we synthetically generate a dataset $D$ which contains a single sample with a fingerprint. Importantly, we remove the non-augmented version of that sample from $D_p$. In practice, this can be any kind of fingerprint that appears only once in the entire training dataset. To ease the training of identification classifiers, we choose a grey constant circle with a radius of 4 pixels but

we also experiment with realistic fingerprints in Sec. 5.4. Therefore, the SAF sample $\mathbf{x}_p$ is defined as an augmented version of a real sample:

$$\mathbf{x}_p = \mathbf{x}_i * (1 - L_p) + \mathbf{x}_{SAF} * L_p, \tag{9}$$

where $i$ is a randomly drawn sample from $D$. The location of the SAF determined by $L_p$ is randomly chosen to lie entirely within the boundary of the image. Then we train a score-based model $\mathbf{s}_\theta(\mathbf{x}, \sigma_t)$ on the augmented dataset $D_p$. To quantify whether or not the trained model is privacy concerning, we define an adversarial attacker that knows of the fingerprint $\mathbf{x}_{SAF}$ and that we can train on detecting this fingerprint. We will refer to this classifier as $c_p(\mathbf{x})$. The second classifier $c_{id}(\mathbf{x})$ is trained on $D$ in a one-versus-all approach to classifying the image's identity. We assume that private information is given away when this classifier correctly detects the generated sample. Importantly, we train $c_{id}(\mathbf{x})$ with random masking using the same circular patches that were used to generate $L_p$. Therefore we can use $c_p(\mathbf{x})$ to filter out all images that contain the SAF and then determine whether or not this sample raises privacy concerns by computing the prediction for $c_{id}(\mathbf{x}')$ generated samples from the generative model $s_\theta$. Theoretically, only one classifier would suffice. This classifier can be interchanged with every other classifier including simple $L_2$-Norm classifiers or more advanced classifiers built with foundation models such as SAM (Kirillov et al., 2023). We choose to split them and focus our investigation on two classifiers with separate goals that can be achieved through strong data augmentation. Furthermore, following our approach we can disentangle the memorization of the SAF from the memorization of $\mathbf{x}_p$.

### 3.4 BOUNDARY COMPUTATION

To give an estimate for $q(p)$ we observe that it only depends on the likelihood $p(\mathbf{x}_p)$ and $t'$, which is supposed to capture the entire region of $\Omega_p$. Therefore, we use $\mathbf{x}'_p$ as input to the classifiers and define $\Omega_p$ as the region where both classifiers give a positive prediction. Since exact likelihood computation and the terms for the variance derived in Eq. (8) reach computationally infeasible value ranges, we can use $t'$ as an indicator of how unlikely it is to generate critical samples from the model. We add a pseudo algorithm for the computation to the supplementary material.

Now, we can freely choose $M$ as parameter and trade-off accuracy for computation time. Given $\mathbf{x}_p$ we define $q_M(p|x_{t,p})$ as the estimate of staying within the boundary of $\Omega_p$ for a given diffusion step $t$. Then we define $t' := \max(\mathbb{T})$, $\mathbb{T} := \{\forall t: q_M(p|x_{t,p}) > 0\}$.

## 4 RELATED WORK

Generative models have disrupted various fields by generating new data instances from the same distribution as the input data. These models include Variational Autoencoders (VAEs) (Kingma & Welling, 2013), Generative Adversarial Networks (GANs) (Goodfellow et al., 2014), and more recently, Generative Diffusion Models (GDMs). Diffusion models can be categorized as score-based generative models (Song & Ermon, 2019; Song et al., 2020b) and models that invert a fixed noise injection process (Sohl-Dickstein et al., 2015; Ho et al., 2020).

Evaluating data privacy in machine learning has been a longstanding concern (Dwork et al., 2006; Abadi et al., 2016; van den Burg & Williams, 2021). Research on integrating privacy-preserving mechanisms in generative models is still in its infancy. Xie et al. (2018) proposed a method to make GANs deferentially private by modifying the training algorithm. Jiang et al. (2022) applied differential privacy to VAEs, and Dockhorn et al. (2022) applied it to diffusion models, showcasing the possibility of explicitly integrating privacy preservation into generative models.

Despite the progress in privacy-preserving generative models, little work has been done on evaluating inherent privacy preservation in diffusion models and providing privacy guaranteed dependant on the training regime. To the best of our knowledge, our work is the first to investigate natural privacy preservation in generative diffusion models, contributing to the ongoing discussion of privacy in machine learning.

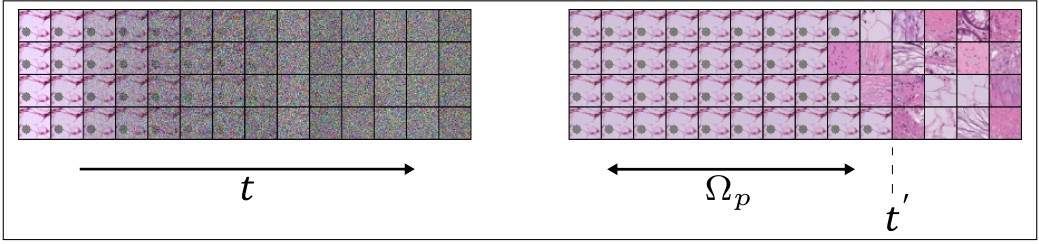

Figure 2: Illustration of the reverse diffusion process. Left shows query images $\mathbf{x}_{t,p}$ for $t \in [0, 0.7]$. Right shows the resulting sample.

## 5 EXPERIMENTS

### 5.1 DATASET

For our experiments we use MedMNISTv2 (Yang et al., 2021). This dataset consists of a combination of multiple downsampled $28 \times 28$ images from different modalities. For single-channel images we repeat the channel dimension three times. For our main experiments we choose PathMNIST, due to the high amount of samples available for that dataset. Furthermore, we experiment with an a-priori selected selection of modalities from this dataset which ranges through multiple sizes and multiple channels of the dataset.

### 5.2 MODELS

The classifiers are randomly initialized ResNet50 (He et al., 2016) architectures. To maximize robustness we employ AugMix (Hendrycks et al., 2020) and in the case of $c_{id}(\mathbf{x})$ we inject random Gaussian noise into the training images to increase the robustness towards possible artifacts from the diffusion process. Furthermore, we randomly mask out patches of the same shape as the SAF to reduce the effect of SAF on the prediction. The training and sampling of the score-model follows the implementation of Song et al. (2020b) with sub-VP SDE sampler due to their reported good performance on exact likelihood computation (Song et al., 2020b) with a custom U-Net architecture based on von Platen et al. (2022). Training $s_\theta$ is done on a single A100 GPU and takes roughly eleven hours. The classifiers are trained until convergence with a validation error patience of 20 epochs, which takes less than one hour. Exhaustive search for $t'$, which is done by computing $q_{M=16}(p|x_{t,p})$ for all $t \in 0, \ldots, 1$, takes four hours.

### 5.3 REVERSE DIFFUSION PROCESS

First, we want to investigate the influence of the size of the dataset on its memorization capabilities. Therefore, we train models on different $|N_D|$ and observe the influence on memorization. We keep the number of training steps fixed at 30000 steps because we observed that this is the length it takes the model to learn to reproduce samples for the smallest subset. After training we sample 150000 images for every model and measure the probability of reproducing our sample at test time. We do this by defining the null-hypothesis $H_0$ that the probability of sampling $\mathbf{x}_p$ is equal to $1/N_D$. Hypothesis $H_1$ claims that the probability is lower. Therefore, we sample 150000 images for every trained model with dataset size $|N_D| \in \{1000, 5000, 10000, 20000, 50000\}$. The results are shown in Tab. 1 It can be seen that the model only learned to reproduce samples with the SAF when the dataset size was comparably low. For $|N_D| = 1000$ the model was surprisingly close to the expected value, indicating that the size of the data is too small relative to the available parameter space and the model memorizes them as discrete distribution of 1000 unrelated images. Every other model produces very few positive predictions from the classifier all of which turn out to be false positives.

The combined prediction $q := c_{id}(\mathbf{x})^+ \cap c_p(\mathbf{x})^+$ is only positive for the smallest dataset. All the larger models do not have any positive samples in their dataset. The p-value for this is smaller than 5% in all cases, meaning that we can reject the null-hypothesis and assume that the probability of $\mathbf{x}_p$ is smaller. Next we look at the samples of different sizes. Initial observation suggest that image quality drops for medium-sized datasets. However, upon closer inspection we see that the

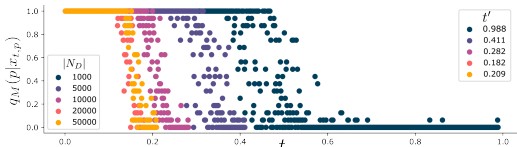

Figure 3: Likelihood of producing $\mathbf{x}_p$ at sampling time as a function of $t$ for $t \in \{0, \ldots, 0.5\}$ and $M = 16$. We stop plotting probabilities after $t'$. Due to the high observed probabilities of the $N_D = 1000$ model, we also compute and plot the probabilities for higher t.

| $|N_D|$ | 1000 | 5000 | 10000 | 20000 | 50000 |
|---|---|---|---|---|---|
| $\mathbb{E}\left[|q|\right]$ | 150 | 30 | 15 | 7.5 | 3 |
| $|c_p(\mathbf{x})^+|$ | 151 | 0 | 0 | 1 | 1 |
| $|c_{id}(\mathbf{x})^+|$ | 151 | 0 | 3 | 3 | 4 |
| $|q|$ | 151 | 0 | 0 | 0 | 0 |

Table 1: Number of positive predictions of the classifiers for models trained on different dataset size on 150000 images. All models use the same classifiers.

| Description | | SAF Classification | | Data Synthesis | | | | |
|---|---|---|---|---|---|---|---|---|
| Dataset | $|N_D|$ | SAF (%) | ID (%) | $\mathrm{FID}_{train}$ | $\mathrm{FID}_{test}$ | $\mathbb{E}(|q|)$ | $|q|$ | $t'$ |
| RetinaMNIST | 1080 | 100 | 99.6 | 5.9 | 19.7 | 46.3 | 52 | 0.998 |
| BloodMNIST | 11959 | 100 | 99.5 | 9.3 | 11. | 4.2 | 0 | 0.241 |
| ChestMNIST | 78468 | 99.93 | 99.8 | 3.3 | 3.9 | 0.6 | 0 | 0.206 |
| PneumoniaMNIST | 4708 | 100 | 99.8 | 9.5 | 28.4 | 10.6 | 2 | 0.719 |
| BreastMNIST | 546 | 100 | 98.7 | 9.2 | 62.6 | 91.6 | 57 | 0.886 |
| OrganSMNIST | 13940 | 99.47 | 99.8 | 19.6 | 19.7 | 3.6 | 0 | 0.582 |

Table 2: Training results for different MedMNIST datasets. We report test accuracy for the SAF classifier but only training accuracy for the ID classifier as identification only makes sense if the sample was part of the training set. For the generative scores we use 50000 samples.

smallest model simply learns to reproduce training data, which can be seen by the fact that some images appear multiple times. This confirms our observation that the model learned the training distribution in the form a discrete set of 1000 images but never learned to generalize. In the context of data-sharing this would mean that the model is essentially a way of saving and retrieving training data but sharing it would raise major privacy issues. The model trained on 5000 images seems to lie in between generalizing and memorizing the learned distribution but the size of dataset was not large enough to learn a meaningful representation. The result indicate that the model learned low frequency information such as color or larger structure, but the images are lacking detail. We provide qualitative results to the supplementary material of our paper.

Now we can use our proposed estimation method to compute $t'$ for all datasets. M is set to a fixed value of 16. The results are shown in Fig. 3. Clearly, the probability for generating samples $q_M(p|x_{t,p})$ decreases with increasing t. More importantly, the threshold at what point the probability drops, is higher for smaller $|N_D|$, which means $t'$ is indeed an important indicator for $q(p)$. Additionally, these results show that sharing the model with $|N_D| = 5000$ would raise more privacy issues as other models, as the indicator suggests that the probability for a sample being generated at inference time is high.

Finally we validate our results by looking at different datasets in Tab. 2. The results confirm our observations of a high amount of memorization in models with small dataset sizes close to the expected value. There is once again a turning point at around 5000 images where samples are no longer memorized. We can also confirm this gap by comparing the FID by calculating it on the training and test dataset. The drop is large in all cases where training samples are memorized. However, FID fails to measure the extent of this effect. PneumoniaMNIST has a larger drop in performance than RetinaMNIST but barely any memorized samples. Our proposed indicator $t'$ on the other hand captures this observation. Furthermore, it is also lower for the BreastMNIST dataset, which according to the high difference between $\mathbb{E}(|q|)$ and $|q|$, did not collapse as strongly towards only reproducing $\mathbf{x}_p$.

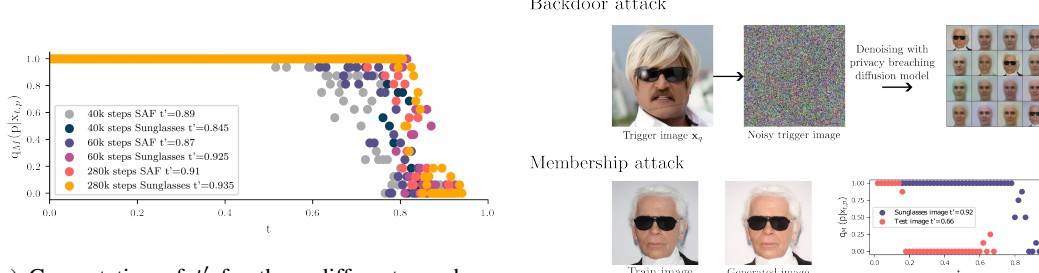

(a) Computation of $t'$ for three different epochs on Celeba-HQ (Karras et al., 2018). We use the SAF as synthetic and sunglasses as non-synthetic fingerprint

(b) Two different attacks that can be quantified using our proposed method.

Figure 4: Scalability experiments to real world datasets.

## 5.4 SCALABILITY ANALYSIS

Next, we investigate whether we can reproduce the results on larger images. Therefore, we employ improved denoising diffusion probabilistic models (Nichol & Dhariwal, 2021) on the CelebA-HQ dataset (Karras et al., 2018) resized to have the size of $256 \times 256$. Switching to diffusion models is not a problem due to the reasonably small discretization error (Su et al., 2023). We scale the size of the SAF proportionally and use a subset $|N_D| = 1000$ following the observations from Tab. 1 to purposefully show memorization for larger images. Additionally, we ensure that the dataset only contains a single image with sunglasses and use this feature as a non-synthetic fingerprint.

Figure 4a shows the computation of $t'$ for three models, one trained for 40000 steps, one for 60000 steps, and the third one trained to overfit on 280000 steps. The results show that $t'$ is generally lower for the 40000 steps model than after 60000 steps. To do a naive search we generate 5000 images and use an SAF classifier and a sunglasses classifier to search for fingerprints in generated samples. The 40000 model did not reproduce the sample, however, the 60000 model reproduced the SAF once. Interestingly, the results match with the non-synthetic fingerprint, where a single image has been reproduced at training time. However, all three models have high $t'$ values indicating that sharing them could be privacy-concerning. Notably, the model trained for 280000 steps did not reveal that it had memorized the training sample.Out of the 5000, none were the sunglass image. Upon close inspection, we observe something similar to mode-collapse as all of the images share similar visual properties (*e.g.*, dark hair).

Overall the results confirm the observation that diffusion models reproduce training images at sampling time and that we can measure this by measuring $t'$ for all three models. We make similar observations on experiments conducted on Stable Diffusion v1.4 Rombach et al. (2022) following the descriptions by Carlini et al. (2023). However, since this model is a text-conditional model, it requires more experimentation. More details can be found in the supplementary material.

Our work is related to backdoor learning attacks where adversaries use the trained model to inject images into the diffusion process to generate inappropriate images (Chou et al., 2023). In our case, an inappropriate image would be a privacy breach. The attack works by exchanging the query image $\mathbf{x}_p$ with a trigger image $\mathbf{x}_q$ that was not part of the training set and that shares visual similarities. In Figure 4b, we show that this attack can be used to increase $q_M(p|x_{t,q})$ of $\mathbf{x}_p$ being generated to $18.75\%$. Since $t'$ is related to the variance of the change of the image performed through the diffusion model according to Eq. (1), we can measure the model's susceptibility to this attack. Sharing the model would be safe if $t'$ is small enough so that the change of the variance is too small to change the images from the trigger to the target image. Additionally, we test if we can use $t'$ to infer information about the membership of an image in the training dataset. Figure 4b demonstrates notably lower $t'$ values for training set images than test set images. Designing a backdoor attack also confirms our observations from 4a as all three models were susceptible to it and reproduced the training image, even the one where we observed mode collapse. This underlines the efficacy of our method, as this case might have been overlooked when using a naive search.

## 6 DISCUSSION

We have shown that $t'$ is a useful indicator towards estimating $q(p)$ since it can be directly derived from it as shown in Sec. 3.1. Our results show that training and publishing trained models without care can lead to critical privacy breaches due to direct data-sharing. The results also suggest that SAFs are either memorized or ignored. This has important implications on the feasibility of using these models instead of direct data sharing as this impedes the ability to use the shared model for datasets with naturally occurring anomalies. These features are vital for medical applications. However, they are rarely replicated during sampling, which further complicates their detection in subsequent applications like anomaly detection.

Computation of $t'$ does not necessarily require the existence of $c_p$. It only requires $c_{id}$. Initially, we assumed that SAF might be reproduced on new synthetic images. Through our approach, we have determined that this is incorrect. The first classifier finds all synthetic instances with the SAF, but if they do not reveal the identity, they are not a problem. They would even be helpful: If the SAF appears in multiple synthetic images, mapping the SAF to the identity would not be straightforward. But in Tab. 1 we can see this is never the case. If the SAF is reproduced, it is always a privacy breach ($|q| == |c_p(\mathbf{x})^+| == |c_{id}(\mathbf{x})^+|$ given that all the positives were false positives, as shown in the supplementary material). This analysis is only possible by splitting $c_{id}$ from $c_p$.

## 7 LIMITATIONS

One limitations of our analysis is that the problem of memorization only occurs for larger images if we fine-tune models on small datasets. However, as Carlini et al. (2023) have shown, this problem also occurs in conditional models, where the marginal probability is much smaller and therefore also the datasets. We show in the supplements that our method even works for conditional methods, but a fair comparison remains challening. We believe the potential for diffusion models is highest when applied to small datasets, especially for data-sharing applications.

Our experiments consider clear synthetic outliers that are not necessarily congruent to the real image distribution. It would be interesting to see if the effect is different if the SAF is closer to the real image. However, the fact that they are visually distinguishable from everything else is necessary for the image to remain detectable and also for the assumption that they pose a privacy concern. Furthermore, because of the problem's high dimensionality, we do not calculate a true probability estimate for the data's occurrence during sampling. Instead, we provide an indicator. The indicator $t'$ has a high variance for large $t$ if $|N_D|$ is small due to the high stochasticity involved when sampling $q_M(p|x_{t,p})$. Therefore, results with $t'$ close to 1 are hard to compare against each other. But as we have shown, these are the cases in which the models raise a privacy concern and direct sampling of $\mathbf{x}_p$ is possible according to Tab. 1. Finally, our derivation of the likelihood misses guarantees for the tightness of the upper bound. We work around this by only computing the indicator which we demonstrate to be useful.

## 8 CONCLUSION

In this work, we have described scenarios in which training score-based models on personal identifiable information like image data can lead to data-sharing issues. By defining an adversarial with prior information about a visual property of the data, we showed that training and publishing these models without care can lead to critical privacy breaches. To illustrate this, we have derived an indicator for the likelihood of reproducing training samples at test time. The results show that generative models trained on small datasets or long training times should not be readily shared. Larger dataset sizes, on the other hand, lead to the model ignoring and never reproducing the detectable fingerprints. We believe enabling the safe training of generative models on small datasets is crucial. Since we can use our method to measure the susceptibility to data memorization, we believe it is also possible to employ it in the training procedure. Developing inherent training techniques has a vast potential to be used for anonymized sharing of private data, which could, in return, improve many problems related to domain adaption or generalization. In the future, we will work on generalizing the idea of $t'$ to different generative models and using it as a direct measure to minimize memorization at training time.

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
