

Figure 5: The accuracy of our estimate on a synthetic two dimensional example. The red dot is the query image $\mathbf{x}_p$ and the circle is the boundary $\Omega_p$ Left: Monte carlo estimation method. Middle: Idealised visualization of our proposed estimation method. Right: Accuracy of both estimation methods.

## A  ESTIMATION OF TIGHTNESS OF BOUND

We show how Monte-Carlo sampling compares to our approach in a two-dimensional synthetic scenario with a bimodal Gaussian distribution shown in Fig. 5. The sampling procedure of our method is shown in the middle. We use an idealized scenario for sampling by choosing values on the exact circle around the query image. Importantly, we see that our proposed method successfully works as an upper bound for the real probability, whereas Monte-Carlo sampling underestimates the real probability at first. Additionally, we see that the estimate is close to the real value and would give a reasonably good estimate from only 32 samples.

## B  MODEL TRAINING DETAILS

To further elaborate on the training details of $c_p(\mathbf{x})$ and $c_{id}(\mathbf{x})$, we show training samples for both classifiers in Fig. 6. Since both tasks are fairly easy binary classification tasks, we employed strong augmentation techniques to ensure that positively predicted samples from the classifiers are SAFs. We balanced the classification task for $c_{id}(\mathbf{x})$ by adding SAFs to 50% of the training images. For validation, we reduce this to 10% to remain closer to the expected distribution. For $c_{id}(\mathbf{x})$ we chose circular masking as training augmentation because we expected it might be necessary to mask out the SAF from the positive predictions of $c_p(\mathbf{x})$. However, closer inspection of the predictions showed this was unnecessary (compare Fig. 7). Another reason is, that we do not want to confuse the model at inference time by showing it SAFs which are not part of the training data of $c_{id}(\mathbf{x})$. The probability of $\mathbf{x}_p$ appearing in the training dataset of $c_{id}(\mathbf{x})$ is set to 10% during training and 50% during validation. The custom diffusion model architecture is based on the open-source implementation of a 2D U-Net[1]. Due to the $28 \times 28$ input images we are forced only to use the three outermost downsampling and upsampling layers.

## C  ESTIMATION ALGORITHM

In Alg. 1 we describe our proposed algorithm to compute the indicator t'. To do an exhaustive search we set the step size to be the same as the sampling step size, start from the maximum value, and go to the minimum value. Since this computation takes too long to be feasible, we experiment with increased step sizes. To improve the computation time even further it is straightforward to change the algorithm to a binary search version or to increase the sampling step size.

## D  FALSE POSITIVE PREDICTIONS OF $c_{id}(\mathbf{x})$ AND $c_p(\mathbf{x})$

The trained models only produce up to five false positives for 150000 generated images, as discussed in Section 5.3. The false positives for all $|N_D|$ are shown in Fig. 7. Both misclassified samples

---

[1] https://github.com/huggingface/diffusers

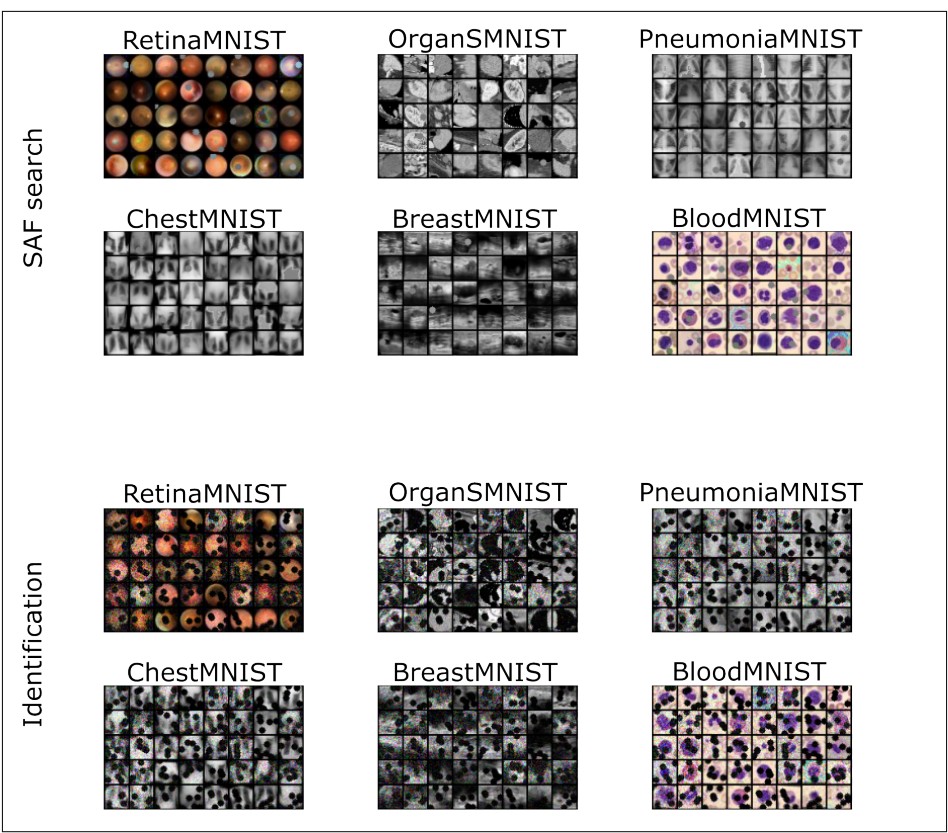

Figure 6: Training image samples for $c_p(\mathbf{x})$ and $c_{id}(\mathbf{x})$

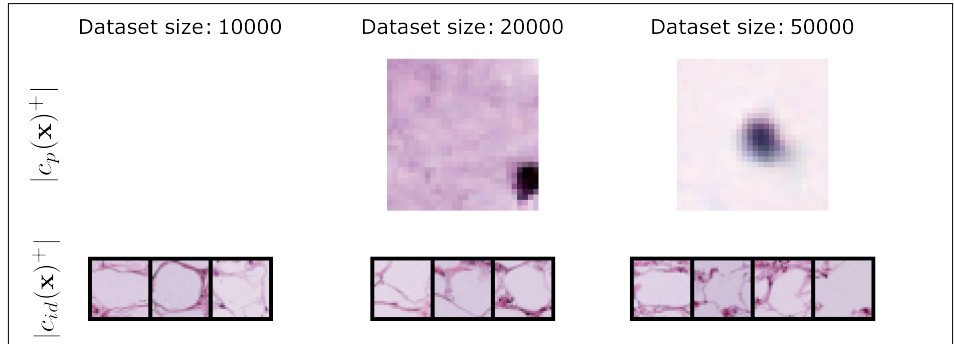

Figure 7: All false positive predictions from the 750000 generated images. All misclassified images by one classifier were filtered and correctly classified by the other classifier.

**Input:** $M, s_\theta(\mathbf{x}, t), c_{SAF}(\mathbf{x}), c_{ID}(\mathbf{x}), \mathbf{x}_p$
**Result:** $t'$

**for** $t = 1, \ldots, 0$ **do**
    **for** $m = 1, \ldots, M$ **do**
        $\mathbf{x}_{t,p} = p(\mathbf{x}_t \mid \mathbf{x}_p)$
        **for** $\tilde{t} = t, \ldots, 0$ **do**
            $\mathbf{x}'_{t,p} = s_\theta(\mathbf{x}'_{t,p}, \tilde{t})$
        **end**
        $\mathbf{x}'_p = \mathbf{x}'_{t,p}$
        **if** $c_{SAF}(\mathbf{x})$ *is True* **and** $c_{ID}(\mathbf{x})$ *is True* **then**
            **return** t
        **end**
    **end**
**end**

**Algorithm 1:** Upper bound likelihood estimation algorithm

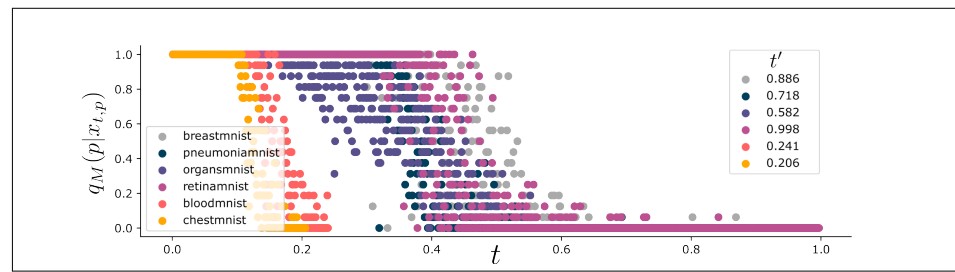

Figure 8: Likelihood of producing $\mathbf{x}_p$ at sampling time as a function of $t$ for $t \in \{0, \ldots, 1\}$ and $M = 16$. We stop plotting probabilities after $t'$.

from $c_{id}(\mathbf{x})$ show great resemblance to the SAF by consisting of a circular monochrome patch. The misclassified identification samples are really similar in terms of texture, color, and structure, although the differences to $\mathbf{x}_p$ are distinct. None of the $c_{id}(\mathbf{x})^+$ would lead to clear privacy issues in practice, which we successfully capture by computing $|q| = 0$ for these three models.

## E    DETAILED RESULTS ON OTHER DATASETS

Next, we report the detailed results for other MedMNIST datasets. This time we perform an exhaustive search for $t'$ and visualize the results in Fig. 8. The trained generative models exhibit the same behavior of starting a slow decline in the probability of reproducing training samples. The end of the decline can be estimated by computing $t'$.

## F    MAE OF MEMORIZED TRAINING SAMPLES

Our pipeline unveiled that training the score-based generative model for a long time on a small dataset leads to reproducing images at sampling time. We show this by applying our classification pipeline and filtering out all negative samples to get $q$. Fig. 9 shows how much these samples are memorized. As can be seen, the sampled images $\mathbf{x}'_p$ are barely distinguishable from the training image $\mathbf{x}_p$. Interestingly, the mean squared error (MSE) between these images goes down rapidly but seems to stagnate after 19000 steps, at which point the reconstruction does not improve much, despite the observed higher memorization probability $q$ reported in Chapter 5.3. This suggests that overfitting occurs not only in the last reverse diffusion steps but also for higher $t$.

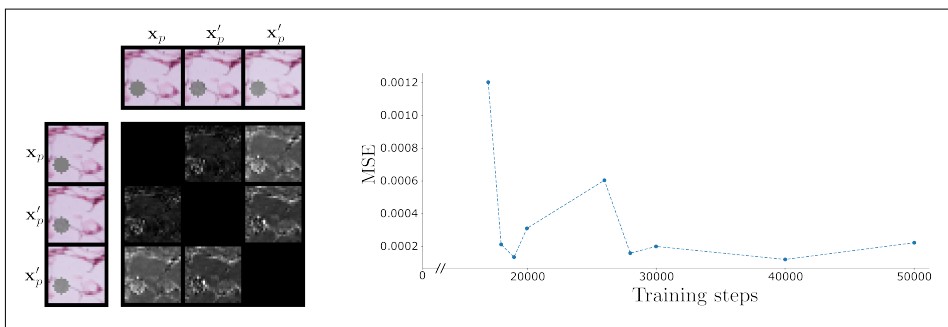

Figure 9: The figure shows a grid-wise comparison of absolute pixel error between the training image $\mathbf{x}_p$ and two sampled image $\mathbf{x}'_p$ that raise privacy concerns (left) and the mean squared error (MSE) for an increasing amount of different training steps (right). $|N_D|$ is set to 1000. The samples on the left are from the model trained for 17000 steps.

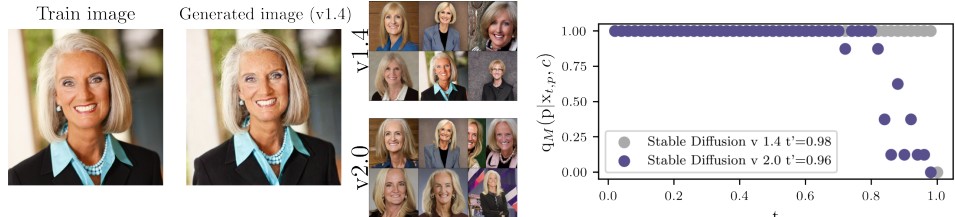

Figure 10: The problem of memorization: Conditional diffusion models memorize training data. Left: shows training and generation examples for both models. SDv1 Rombach et al. (2022) reproduces training samples (reproduced from Carlini et al. (2023)). Version 2 of of the same model no longer exhibits this problem. Right: We show that using our proposed method we can measure this.

## G   RESULTS ON STABLE DIFFUSION

Reproduces the privacy problems of Stable diffusion v1.4 (Rombach et al., 2022) which were first discovered by Carlini et al. (2023). We prompt a text conditional model on a name and see that it reproduces the training image at sampling time in one out of sixteen cases. Interestingly we did not observe this for Stable diffusion v2.0, which is a fine-tuned version of the same model. Using our proposed method, we can measure this. Equation 1 can be extended to conditional models. Therefore, we train a single classifier on re-identification of the image by using 500 randomly selected images of the same person generated by Stable diffusion v2.0. The results are shown in Fig. 10 and show that we can quantify this difference in memorization which underlines that our method is useful in practice and even can be applied to pre-trained models.

## H   TRAINING LENGTH

We experiment with the influence of the training length on $|p|$ by sampling 10000 images from a model trained on $|N_D| = 1000$ and show the results in Fig. 2. For the first 14000 steps, the model only learns low-frequency attributes of the data. The visual quality is low and therefore also the probability of reproducing $\mathbf{x}_p$. Around 20000 the quality of the generated samples improves visually, but also the number of memorized training samples. At this point, the model already starts to accurately reproduce $\mathbf{x}_p$ at sampling time. Every detected sample is visually indistinguishable from the training image. The MAE even goes down to $1 \times 10^{-4}$. Based on these observations, we continue our investigations with a fixed training length of 30000 steps.

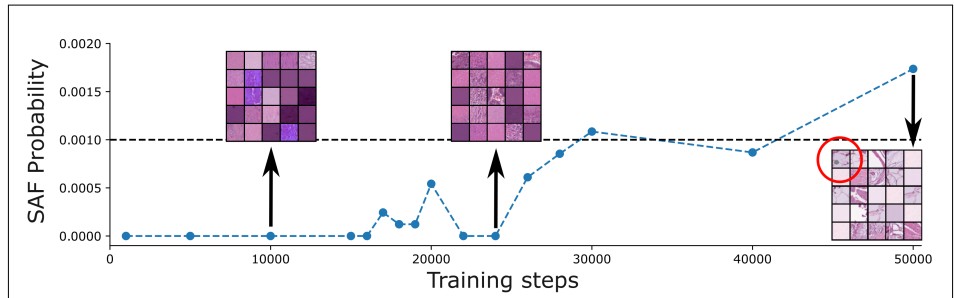

Figure 11: Influence of training length on generative and memorization properties. A positively classified sample can be seen in the top-left corner of the rightmost image.

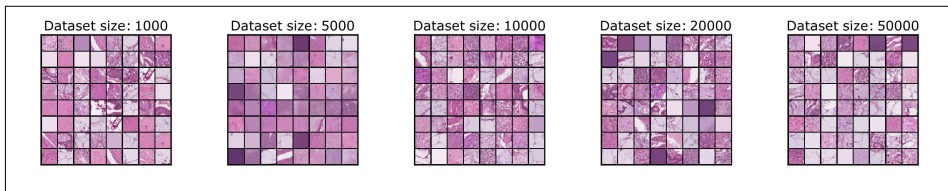

Figure 12: Representative samples from trained models on different dataset sizes $|N_D|$.

# I  DATASET SIZE

Fig. 12 shows visual results of training the same diffusion model on different dataset sizes. As shown in Fig. 3, the first model merely memorizes the samples, while the last model learns the underlying distribution and generalizes. This is nicely captured by t'.