# OpenReview forum: "Quantifying Anonymity in Score-Based Generators with Adversarial Fingerprinting"
_ICLR.cc/2024/Conference — Submitted to ICLR 2024_

### Official Review · Reviewer_Kiwu · 2023-10-31

**Soundness:** 3 good
**Presentation:** 2 fair
**Contribution:** 3 good
**Rating:** 5
**Confidence:** 3

**Summary:**

This paper focuses on the possible privacy breach concerns when sharing generative models. To investigate the possibility of leaking privacy in a generative model, this paper proposes an approach that can estimate the maximum probability of sensitive data being reproduced by the adversary with a specific fingerprint. Experimental results show that privacy breaches could happen when the generative models are trained on small datasets or with long training times.

**Strengths:**

1. The proposed method for estimating the maximum probability of a sample being reproduced is conceptually easy to follow and reasonable.
2. The findings on the conditions for when generative models are prone to privacy breaches are plausible, and this seems to be attributable to overfitting.
3. Some classic attack methods, e.g., backdoor and membership inference, can be quantified using the proposed method.
4. The experiments can evaluate the proposed method well,  which looks sufficient and convincing.

**Weaknesses:**

1.  The experimental results are mainly conducted on the datasets from the MNIST list. Please add some other datasets to evaluate the effectiveness of the proposed method. Does the proposed method exclusively pertain to healthcare? If so, I do not see the relationship between the method and the healthcare scenario.
2. There are a lot of attack methods not only limited to backdoor and membership inference, e.g., feature inference, model inversion, hijack, model steal, etc. This paper aims to quantify the privacy risk. However, some attacks are not privacy issues, i.e., backdoor, hijack and model steal attacks. The authors should investigate the relationship between attack methods and the proposed method.
3. The authors should analyze the relationship between attack methods and the proposed method from both theoretical and experimental aspects. Every attack method has its own characteristics.
4. Flaws in writing: there are some flaws in the writing, e.g., lacking a “.” at the end of the second point of contribution and the first paragraph of the background section.
5. Figure 2 should be referred to in the text of this paper to help readers understand.

**Questions:**

1.  Please include additional datasets to assess the effectiveness of the proposed method. Does the proposed method exclusively pertain to healthcare? If so, I do not see the relationship between the method and the healthcare scenario.
2. This paper aims to quantify the privacy risk. However, some attacks are not privacy issues, i.e., backdoor, hijack and model steal attacks. The authors should investigate the relationship between attack methods and the proposed method.
3. The authors should analyze the relationship between attack methods and the proposed method from both theoretical and experimental aspects. Every attack method has its own characteristics.
4. Flaws in writing: there are some flaws in the writing, e.g., lacking a “.” at the end of the second point of contribution and the first paragraph of the background section.
5. Figure 2 should be referred to in the text of this paper to help readers understand.

---

### Official Review · Reviewer_RsRi · 2023-10-31

**Soundness:** 3 good
**Presentation:** 2 fair
**Contribution:** 2 fair
**Rating:** 3
**Confidence:** 4

**Summary:**

This paper explores the idea of using diffusion models trained on private data as a data-sharing strategy for sensitive contexts such as healthcare and proposes a method to quantify the anonymity of their sampling process. The paper proposes a method to evaluate the memorization of training data by estimating the upper bound of the likelihood of reproducing sampling from the entire subspace of samples similar to the target sample. The authors evaluate their method with synthetic anatomic fingerprints in the form of 4px-radius gray circles used to augment several MedMNIST datasets and sunglasses in the Celeba-HQ. The experiments show that generative models trained on small datasets are more susceptible to memorization.

**Strengths:**

The use reverse diffusion process to estimate the subspace of sample similar to the target sample is original.

**Weaknesses:**

The main limitation of the paper is the evaluation of the proposed solution. While the motivation indicates realistic fingerprint (e.g., skin tattoo, implant, heart monitor), the experiments consider simple scenarios (gray circles, unique sunglasses), which do not help appreciate the true capability of the solution. Additionally, the paper appears to use lack of memorization and lack of privacy risks interchangeably. There are other forms of privacy risks, such as attribute inference or property inference, that could still prevent the sharing of generative models even if memorization is addressed. I suggest authors clarify that aspect in the paper.

**Questions:**

- In a realistic setup, there could be different variants of the same fingerprint on multiple samples. How would the solution perform in such a case?

---

### Official Review · Reviewer_kVLX · 2023-11-04

**Soundness:** 2 fair
**Presentation:** 2 fair
**Contribution:** 1 poor
**Rating:** 3
**Confidence:** 4

**Summary:**

This paper investigates the potential privacy issues associated with score-based generative models, specifically by introducing a method for estimating the (upper bound of the) probability that identifiable training images will be reproduced during the sampling/generation process. Experimental results validate the concern that images with identifiable fingerprints can be reproduced at sampling time if the models are trained without appropriate privacy safeguards.

**Strengths:**

- Investigating potential privacy issues associated with the use of generative models is an interesting topic and is of practical value.

- The idea of estimating regeneration probability using likelihoods derived from the training objective and generation process should be a valid and promising approach that aligns well with the principles of score-based generative modeling.

**Weaknesses:**

1. The key contributions of this submission are somewhat unclear:
   - The main finding—that recent generative models are likely to compromise privacy and potentially copyright—has been discussed in earlier publications, some of which are cited in this submission. This observation raises questions about the novelty of this finding. Furthermore, the scenario presented, which focuses on "anomalies", does not seem to represent an improvement over existing ones in terms of practical relevance. Stronger results have been reported in the literature, indicating that even neutral-looking images, not just anomalies, could be memorized and reproduced by these models.
   - Additionally, if the authors would like to claim that their proposed approach is a significant contribution, they need to provide a detailed comparison with existing methods. This should include a clear description of its application scenarios (see point 2 below).
Specifically, the submission could address application examples such as membership inference and backdoor attacks, and demonstrate how the proposed approach differs from or improves upon existing methods. To the best of my knowledge, there are indeed numerous studies (>5) on these scenarios (which also includes some likelihood-based estimations), offering approaches that may even have broader applicability (e.g., applicable to general generative models, not just the score-based models discussed here).

2. The potential application scenarios (i.e., the threat model) of the proposed approach are not clearly stated. This ambiguity further negatively affects my assessment of the submission’s contributions. For example, it would be necessary to state how the approach would be used in practice (by a defender or an attacker) and which knowledge (about the model, training configuration, data distribution, etc.) is required for executing the method.

**Questions:**

- The definition of 'private region' using an additional classifier seems to be either non-rigorous or too computationally intensive in practice. Classifying each individual ID would quickly become infeasible for relatively large datasets. Moreover, other more scalable methods might conflict with the proper definition of privacy violations, since simply generating similar-looking images through generalization or hallucination should be permissible. Conversely, for small-scale datasets, the value of the study to the community might be questioned, considering that memorization and overfitting are widely recognized issues.

- The structure of the presentation in the methods section could be further refined to improve the logical flow within subsections 3.1-3.4. Specifically, including a pseudocode representation of the overall pipeline within the main paper could greatly aid understanding.

-  Section 3.4: the meaning of $M$ and $q_M(p|x_{t,p})$ seem not explained and how would these be computed/selected seems missing from the main paper. However, I found this quite critical for understanding the overall approach.

- Section 2: directly stating $p_{\sigma_1}(\tilde{x} | x) ∼ p_{data}(x)$ without constraining $\sigma_1$ looks not rigorous

---

### Meta-Review · Area_Chair_JUd4 · 2023-12-01

**Metareview:**

The paper studies how adversarial approaches to black box models, score based generative models, can reveal fingerprints in images. It introduces a method for estimating the upper bound of the probability of reproducing identifiable training images during the sampling process.

The main weaknesses of the paper are the lack of originality and comparison with the state of the art, numerous methods have been proposed in the literature and the paper should better describe the attack model and scenarios; the evaluation is clearly very limited; the writing should also be polished.

The authors did not respond to the reviewers.

**Justification For Why Not Higher Score:**

The authors did not respond to the reviewers.

**Justification For Why Not Lower Score:**

N/A

---

### Decision · Program_Chairs · 2024-01-16

Reject